# Exploring Spatiotemporal Relations between Soil Moisture, Precipitation, and Streamflow for a Large Set of Watersheds Using Google Earth Engine

**Nazmus Sazib [1,2,*], John Bolten [1] and Iliana Mladenova [3]**

[1] Hydrological Sciences Branch, NASA Goddard Space Flight Center, Greenbelt, MD 20771, USA; john.bolten@nasa.gov

[2] Science Application International Corporation (SAIC), Lanham, MD 20706, USA

[3] United States Department of Agriculture, Washington, DC 20250, USA; Iliana.Mladenova@usda.gov

[*] Correspondence: nazmus.s.sazib@nasa.gov; Tel.: +1-301-614-6384

**Abstract:** An understanding of streamflow variability and its response to changes in climate conditions is essential for water resource planning and management practices that will help to mitigate the impacts of extreme events such as floods and droughts on agriculture and other human activities. This study investigated the relationship between precipitation, soil moisture, and streamflow over a wide range of watersheds across the United States using Google Earth Engine (GEE). The correlation analyses disclosed a strong association between precipitation, soil moisture, and streamflow, however, soil moisture was found to have a higher correlation with the streamflow relative to precipitation. Results indicated different strength of the association depends on the watershed classes and lag times assessments. The perennial watersheds showed higher coherence compared to intermittent watersheds. Previous month precipitation and soil moisture have a stronger influence on the current month streamflow, particularly in the snow-dominated watersheds. Monthly streamflow forecasting models were developed using an autoregressive integrated moving average (ARIMA) and support vector machine (SVM). The results showed that the SVM model generally performed better than the ARIMA model. Overall streamflow forecasting model performance varied considerably among watershed classes, and perennial watersheds tend to exhibit better predictably compared to intermittent watersheds due to lower streamflow variability. The SVM models with precipitation and streamflow inputs performed better than those with streamflow input only. Results indicated that the inclusion of antecedent root-zone soil moisture improved the streamflow forecasting in most of the watersheds, and the largest improvements occurred in the intermittent watersheds. In conclusion, this work demonstrated that knowing the relationship between precipitation, soil moisture, and streamflow in different watershed classes will enhance the understanding of the hydrologic process and can be effectively utilized in improving streamflow forecasting for better satellite-based water resource management strategies.

**Keywords:** soil moisture; streamflow forecasting; Google Earth Engine

## 1. Introduction

Streamflow is an important hydroclimatic variable, which is influenced both by change in climate (e.g., precipitation, temperature) condition and by human activities, including land-use changes, and water use by the agricultural and industrial sectors. Wet soil moisture conditions result in overland flow and possible flooding during an extreme precipitation event [1]. On the contrary, dry soil moisture conditions amplify the occurrence of temperature extremes [2]. Therefore, changes in climate result in changes in hydrologic process which lead to changes in the magnitude and frequency of extreme

hydrologic events [3,4]. Thus, it is of great scientific and practical importance to analyze and quantify the effect of climate-related drivers on streamflow for improved implementation of sustainable and efficient management of water-related systems.

A considerable amount of research has been conducted regarding streamflow response to precipitation changes. Zhao et al. (2009) evaluated the relationship between precipitation and streamflow using sensitivity and simulation-based methods over the yellow river basin in China, where they found that the changes of streamflow are more sensitive to precipitation than evapotranspiration [5]. Hodgkins et al. (2011) investigated the summer base flow and storm flow in the New England region of the United States. They found an increase in summer storm flows by 50% due to a significant increase in summer precipitation [6]. In the specific context of the soil moisture-streamflow relationship, several studies have documented to enhance the understanding of the influence of soil moisture on streamflow generation. For instance, Maurer and Lettenmaier et al. (2003) explored the potential of climatic indicators and the initial condition of simulated snow and soil moisture for runoff predictability throughout the Mississippi River and found that soil moisture is one of the controlling sources of runoff predictability in all seasons [7]. Wang et al. (2018) explored the effects of land use and topography on streamflow and soil moisture response to precipitation. They reported quicker streamflow response to rainfall in the case of watersheds coupled with forests and steep topography compared to pasture and flat topography [8]. In another study, soil moisture was identified as a critical component in the development of the flood warning system [9].

The streamflow-hydroclimatic variables relationship discussed above is also associated with streamflow forecasting, which provides vital information for environmental impact assessments, agriculture studies, climate change impacts, groundwater assessment, and reservoir operations [10,11]. A wide variety of physically-based and data-driven models exist, including autoregressive moving average (ARMA), linear regression (LR), wavelet transform (WT), artificial neural networks (ANNs), support vector machines (SVMs), and also their combinations are commonly used for hydrologic application [12–18]. The ARIMA model utilizes correlation and trends in the historical time series data for forecasting and has been widely used for streamflow forecasting due to easy development and implementation. The ARIMA model has been successfully applied in multiple hydrologic modeling applications, including predicting the streamflow [19,20], rainfall [21,22], and groundwater [19,23]. The ARIMA model and its derivatives, such as seasonal ARIMA (SARIMA), periodic ARIMA, and ARMAX, are applied extensively in streamflow forecasting, particularly in the modeling of monthly streamflow [24–26]. However, the ARIMA model is not efficient in capturing nonlinearity of hydrologic applications due to its underlying assumption that input data have to be normally distributed and stationary and may not always perform well [27]. Therefore, machine learning techniques such as ANN and SVM have gained considerable attention in streamflow forecasting studies due to their ability to identify complex relationships between input and output data sets while inherently handling nonlinearity and non-stationarity of the systems [28].

ANN are black box models that use a transfer function to identify the non-linear relationship between the input and outputs and do not require detailed knowledge of the internal process of a system. Several studies reported the application of ANN on monthly, weekly, and daily streamflow forecasting [29,30] and confirmed the better performance of the ANN model over the traditional statistical techniques in modeling [31,32]. Demirel et al. (2009) compared the performance of the Soil and Water Assessment Tool (SWAT) and ANN for streamflow forecasting and found that the ANN model performed better than the SWAT when forecasting peak streamflow [33]. However, ANN requires a large amount of training data for model development and proper identification of the network (e.g., number of nodes, hidden layers) to overcome the overfitting problems [34]. SVM's relatively new form of machine learning method was developed specifically for the classification and pattern recognition problem and later adopted for the regression analysis. In contrast to other machine learning methods, the support vector machine implements the structural risk minimization

principle (SRM) rather than the empirical risk minimization principle that reduced the overfitting of the model [35].

A number of studies have demonstrated the potential of support vector machine (SVM) in the various hydrological applications and streamflow forecasting. Kalra et al. (2009) applied SVM for streamflow prediction in the Upper Colorado River Basin in the western United States and found the better performance of the SVM when compared with the prediction using artificial neural network and linear regression [34]. Kisi et al. (2015) compared ARMA, least-square support vector regression, and adaptive neuro-fuzzy model to forecast monthly streamflow in the Dicle Basin of Turkey. They reported better performance of the SVM compared to ARMA [36]. Milad et al. (2014) applied the SVM and a physically-based model to predict monthly streamflow over an arid region in the southern part of Iran and found that the SVM monthly predictions were closer to the observed streamflow than the physically-based model [37].

Most of the studies mentioned above were carried out on a particular number of the watershed located in the similar climate condition, thus provide little insight into the performance of those forecasting models on different climate regimes. In addition, those analyses require distinct geospatial and time series data, which can be gathered from different sources such as United State Department of Agriculture (USDA) for geospatial data and the National Oceanic and Atmospheric Administration (NOAA) for climate data. For an individual watershed, this may not be too difficult, but for a vast majority of watersheds, structuring and providing data for each watershed can become a daunting task. Therefore, the inherent heterogeneity, diversity, and abundance of these data make it challenging to follow these approaches for a large number of watersheds, as we do in this study. To this end, a majority of the streamflow forecasting studies used either antecedent streamflow, precipitation, or both as a suitable predictor, and a limited number of studies [38,39] investigated the potential of soil moisture for streamflow forecasting on a large number of watersheds.

Here, we explore the utility of strategically applying precipitation and streamflow for forecasting streamflow over a large number, i.e., 601, watersheds in varying climates, land cover, and terrain. The objectives of this study are: (1) to evaluate the possible linkage among precipitation, soil moisture, and streamflow and (2) to analyze the potential of satellite-based soil moisture for streamflow forecasting over a large set of watersheds across the United States. The manuscript is organized as follows: first, a description of the data and methodology is provided; the analysis results are then presented, and finally, key findings are summarized.

## 2. Materials and Methods

### 2.1. Study Area and Data Used

This study was conducted over 601 watersheds selected from Geospatial Attributes of Gages for Evaluating Streamflow (GAGE) database [40]. The 601 watersheds were selected based on their limited anthropogenic impact, i.e., development and hydropower management. The GAGE database provides geospatial information for 6785 watersheds across the U.S and identifies 1512 watersheds with minimal human-influence. We selected 601 watersheds out of 1512 watersheds with no missing streamflow data for the period of 2010–2017. Falcone et al. (2010) used three criteria to identify watersheds with minimum human influence, (1) a quantitative index of anthropogenic modification within the watershed based on GIS-derived variables, (2) visual inspection of every stream gage and drainage basin from recent high-resolution imagery and topographic maps, and (3) information about man-made influences from USGS Annual Water Data Reports [41].

Meteorological variables were obtained from the Parameter elevation Regression on Independent Slopes Model (PRISM), which provides precipitation, temperature, and dew point temperature at monthly and daily time scales [42]. It uses a statistical model to produce grid estimates of precipitation, temperature, and dew point using point measure of climate data. Daily streamflow data were obtained from the United States Geological Survey (USGS) National Water Information System (NWIS) website

using the 'readNWISdv' function from the USGS' R 'data Retrieval' package. Soil moisture data sets were obtained from the NASA-USDA global soil moisture product, which is available through the Google Earth Engine (GEE) [43]. The NASA-USDA global soil moisture was developed by merging satellite-derived Soil Moisture Ocean Salinity (SMOS) Level 3 soil moisture observations into the modified two-layer Palmer model using the Ensemble Kalman Filter (EnKF) [44–48]. The Palmer model is a simple water balance model that estimates the amount of gained or lost water in the soil profile by tracking the amount of water lost by evapotranspiration and restored by precipitation. The U.S. Air Force 557th Weather Wing (formerly known as U.S. Air Force Weather Agency, AFWA) precipitation, temperature data are used as inputs in the Palmer model [45,46]. GEE links for the PRISM precipitation and SMOS soil moisture data are provided in the Supplementary Materials.

### 2.2. Data Processing

An overview of major methodological approaches applied in this study is presented in Figure 1. We aggregated soil moisture and precipitation data sets to monthly composites and then spatially averaged over the watershed boundaries using Google Earth Engine (GEE). The GEE archives a petabyte of earth observing remote sensing data and includes processing software that enabled us to do extensive geospatial and temporal analysis using high-performance computing resources. Then, we used soil moisture and precipitation data to explore their spatial and temporal variability over 601 watersheds across U.S. Next, the association between soil moisture, precipitation, and streamflow were estimated using Spearman rank correlation for different lag times between the precipitation and soil moisture and observed streamflow. Lag correlation analysis was performed in order to determine the number of antecedent observations that have influences on streamflow forecasting. We used standardized time series of soil moisture, streamflow, and precipitation data while computing statistical association to minimize the seasonal dependence of those variables. To this end, we isolated the impact of soil moisture and precipitation on the observed streamflow, and quantified the streamflow forecasting potential of those variables. We applied the autoregressive integrated moving average (ARIMA) and support vector machine (SVM) regression models (SVR) for streamflow forecasting as follows:

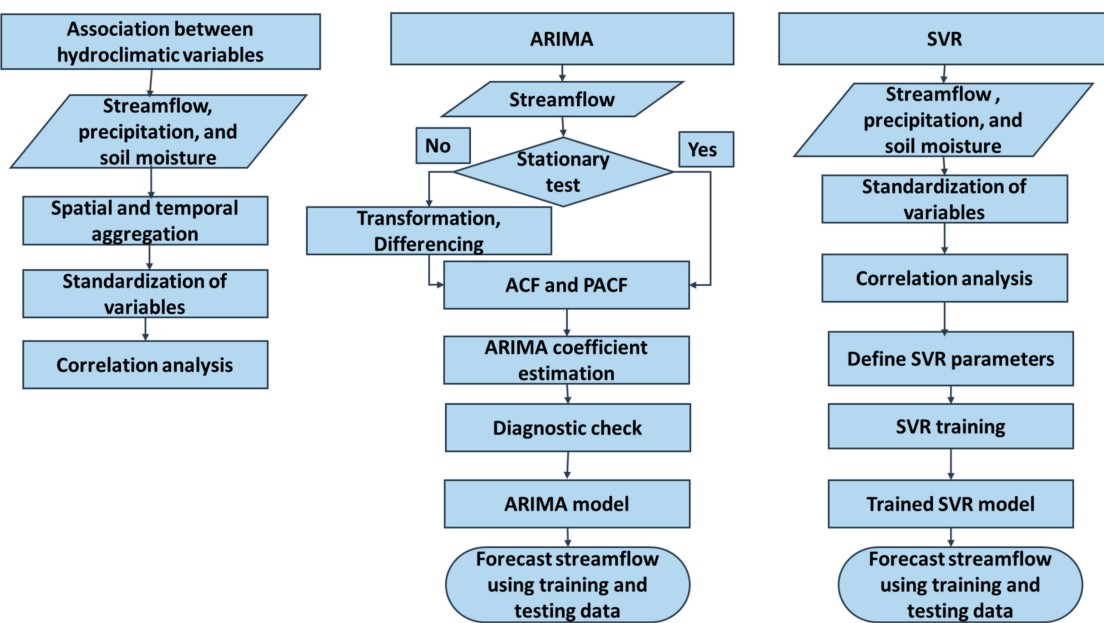

**Figure 1.** Schematic diagram of the methodology used in the study.

Model $Q_{arima}$:

$$Q_t = f(Q_{t-1}, Q_{t-2}, \ldots Q_{t-n,}) \tag{1}$$

Model Q$_{svr1}$:

$$Q_{t=}f(Q_{t-1,}Q_{t-2,}\ldots Q_{t-n,}) \tag{2}$$

Model Q$_{svr2}$:

$$Q_{t=}f(Q_{t-1,}Q_{t-2,}\ldots Q_{t-n,}\quad P_{t-1,}P_{t-2,}\ldots P_{t-n,}) \tag{3}$$

Model Q$_{svr3}$:

$$Q_{t=}f(Q_{t-1,}Q_{t-2,}\ldots Q_{t-n,}P_{t-1,}P_{t-2,}\ldots P_{t-n,} SM_{t-1,}SM_{t-2,}\ldots SM_{t-n,}) \tag{4}$$

Here, $Q_t$ denotes the streamflow at time t and $P_{t-1}, \ldots P_{t-n}, SM_{t-1} \ldots SM_{t-n}$ represent precipitation and soil moisture at time t-1, … t-n respectively. Model Q$_{arima}$ and Model Q$_{svr1}$ were developed using ARIMA and SVR model respectively and consider antecedent streamflow as a predictor. Generally, ARIMA models have been widely used for time series forecasting due to their relative simplicity and effectiveness, however, they are limited by assumptions of normality, linearity, and variable independence [49,50]. The SVR method, which considers the nonlinearity and non-stationary signals in the streamflow, was used in model Q$_{svr1}$, Q$_{svr2}$, and Q$_{svr3}$. Both Q$_{svr2}$ and Q$_{svr3}$ models used antecedent precipitation and streamflow as inputs, and Q$_{svr3}$ included soil moisture as an additional predictor.

## 2.3. Development of ARIMA Model

An ARIMA is a univariate model which utilizes historical time series data to predict future and generally expressed as ARIMA (p, d, and q) where p, d, and q refers to the order of the autoregressive (AR), integrated, (I) and moving average (MA) respectively. The AR component indicates a linear regression model where lagged values of time series are used as predictors for forecasting. The Integrated component refers to the transformation of the non-stationary time series to stationary by performing a d-order differential to the original time series. The MA represents an auto-regression of the residual errors. An ARIMA model with seasonal components also denoted as ARIMA (p, d, and q) (P, D, and Q) m, where m refers to the number of time steps in a season. The uppercase P, D, Q refer to the autoregressive, integrated, and moving average order for the seasonal part of the model. The general formula for the ARIMA model with stationary (d = 0) time series data can be written as follows:

$$Y_{t=}\mu + \varphi_1 Y_{t-1} + \ldots + \varphi_p Y_{t-p} + \theta_1 e_{t-1} + \ldots + \theta_q e_{t-q} \tag{5}$$

where, $Y_t$ represents forecasted streamflow at time t; $Y_{t-1}, \ldots , Y_{t-p}$ denote the streamflow at time t-1, … t-p respectively. $\mu$, and $e_t$ are the constant and white noise; $\varphi$ and $\theta$ are model parameters. The development of an ARIMA model includes three steps: identification, estimation, and diagnostic check. The normality and stationarity of the streamflow data were determined in the identification step, as the inputs of the ARIMA model have to be stationary. The modified Mann-Kendall and Mann-Whitney tests were performed to identify any trend and jump in the monthly streamflow data as those components cause the non-stationary of the time series data [20]. The modified Mann-Kendall method utilized a variance correction approach as proposed by Yue and Wang et al. (2004) to address the issue of serial correlation in the streamflow data [51]. The streamflow data also should have constant variance and normally distributed to meet the stationary criteria. In general, streamflow data are highly skewed; therefore, the box-cox transformation was applied to obtain a homogeneous variance of streamflow data [52]. If there is a seasonality of the data, seasonal differencing was also applied to the monthly data. Next, the autocorrelation and partial autocorrelation analysis were performed on the non-stationary data sets to determine the order of the order of auto-regression (p) and moving average (q). The ACF determines the amount of linear dependence between streamflow data and lags of itself, whereas the PACF identifies the required autoregressive terms to reveal the time lag characteristics [53]. Once the order of the model was identified, the Akaike information criterion

(AIC) was used to determine the optimum model parameter. The AIC estimates the goodness of fit and model parsimony and expressed as:

$$AIC = 2k + Nln\left(\frac{SSE}{N}\right) \tag{6}$$

$$SSE = \sum_{i=1}^{N} \varepsilon_i^2 \tag{7}$$

where k is model parameter, $N$ is the observations number and $\varepsilon$ represents the residual error. The minimum values of the AIC indicate better model performance [54]. In the diagnostic step, the Ljung-Box test was performed to check whether the residuals are independent and normally distributed. Finally, the monthly streamflow was forecasted using the best-fitted model.

## 2.4. Development of SVR Model

Support vector regression (SVR) follows the basic SVM concept and maps the data to a higher-dimensional space, thus, complicated nonlinear relationships between streamflow and other hydrologic variables are maintained and considered. For a given data set, the SVR regression function can be expressed as follows:

$$Y = \omega.\varnothing(x) + b \tag{8}$$

Y and x represent the output and input data, $w$ is a weight vector, $b$ is the bias, and $\varnothing(x)$ presents transfer function. The transfer function uses a nonlinear function to transform input data to a linear model in high dimensional feature space. Thus, it plays a critical role in developing an appropriate regression model. Several transfer functions such as linear, polynomial, and radial basis are available in SVR. In this work, we used the radial basis transfer function as a large number of previous hydrological studies utilized on the forecasting application and reporting satisfactory performance [55,56]. The weight vector can be estimated by minimizing the following regularized risk function:

$$R(c) = \frac{1}{2}\|\omega\|^2 + C\sum_{i=1}^{N}\left(\epsilon_i + \epsilon_i^*\right) \tag{9}$$

Subject to the following condition:

$$\begin{aligned} y_i - (\omega.\varnothing(x_i) + b) &\leq \varepsilon + \epsilon_i \\ (\omega.\varnothing(x_i) + b) - y_i &\leq \varepsilon + \epsilon_i^* \\ \varepsilon, \epsilon_i^* &\geq 0, i = 1, 2, 3, \ldots\ldots.N \end{aligned} \tag{10}$$

Here, C is a user-defined parameter that controls the trade-off between maximizing the margin and minimizing the training error. Higher C value results in overfitting of the model, while the smaller C value may cause the poor approximation of the model. The ξ and ξ* are slack variables that specify the upper and the lower training errors subject to an error tolerance $\varepsilon$, which defines as the difference between observed, and model values calculated from the regression analysis. The solution of equation 9 can be found by using a Lagrangian function and Karush-Kuhn-Tucker complementarity conditions [35,57]. Finally, the SVR-based regression function can be expressed as:

$$Y = \sum_{i=1}^{N}\left(\alpha_i - \alpha_i^*\right).K(x, x_i) + b \tag{11}$$

where, $\alpha_i$ and $\alpha_i^*$ are the Lagrangian multipliers and $K(x, x_i)$ is the radial basis kernel function expressed as:

$$K(x, x_i) = exp\left(-\gamma(\|x_i - x\|)^2\right) \tag{12}$$

where $x_i$ denotes the support vector, $\gamma$ is radial basis kernel parameter which gives the width of the kernel.

SVR model parameters (C, $\gamma$, and $\xi$) were estimated using grid search methods and the range of C and $\gamma$, $\xi$ values were set to (1–100) and (0.01–1) respectively to calibrate the parameter. The number of suitable predictors for the SVR models were also selected based on the auto and cross-correlation analysis [56,58].

Before developing the ARIMA and SVR models, the time series data were separated into two data sets, 80% for training, and 20% for testing. The accuracy of the three streamflow forecasting models was evaluated using Kling-Gupta efficiency (KGE) and root mean square error (RMSE) values, which are defined as follows:

$$KGE = \sqrt{(r-1)^2 + \left(\frac{\sigma_{sim}}{\sigma_{obs}} - 1\right)^2 + \left(\frac{\mu_{sim}}{\mu_{obs}} - 1\right)^2} \tag{13}$$

where, $r$ is the linear correlation between observed and simulated streamflow, $\sigma_{obs}$ and $\sigma_{sim}$ is the standard deviation of the observed and simulated streamflow, $\mu_{obs}$ and $\mu_{sim}$ are the mean observed and simulated streamflow.

$$RMSE = \sqrt{\frac{\sum_{i=1}^{n}(Q_{obs} - Q_{sim})^2}{n}} \tag{14}$$

where, $Q_{obs}$ and $Q_{sim}$ are the observed and simulated streamflow, $n$ is the number of streamflow observation.

KGE can range between $-\infty$ and one, where the value of one indicates a perfect match of forecasting discharge to the observed data. The root mean squared error estimates the mean error between the observed values and the simulated streamflow data. Lower RMSE indicates less error between the simulated and observed streamflow than a large RSME.

Physical characteristics and hydrological fluxes such as streamflow and soil moisture vary across hydrological systems. However, watershed classification allows complexity of the hydrologic system to be classified and organized for a better understanding and conceptualization of the hydrologic process across spatial and temporal scales [59–61]. Therefore, we grouped and discussed our analysis in eight watersheds classes following Dhungel (2016) in lieu of 601 watersheds [40]. The Dhungel et al. (2016) watershed classes were derived from streamflow regime variables that sufficiently describe the spatial and temporal variability of long term streamflow over 601 watersheds. They applied the principal component analysis on sixteen selected streamflow regime variables and identified five major aspects of regime (low-flow, magnitude, flashiness, timing, and constancy) for classification. Then, Ward's hierarchical clustering was applied to the five-streamflow factors to classify the 601 watersheds into eight watershed classes that includes: (A1) Small Steady Perennial, (A2) Large Steady Perennial, (B1) Steady Intermittent, (B21) Early Intermittent, (B22) Late Intermittent, (C1) Early Flashy Perennial, (C21) Small Flashy Perennial, and (C22) Large Flashy Perennial streams. Class A1 and A2 watersheds were distributed mostly in the north mid-western U.S., with larger A2 streams in the northern part, and smaller A1 streams are towards the south (Figure 2). Class B1 watersheds dominated the mostly dry areas of North and South Dakota, and Class B22 watersheds occurred mostly in the central U.S. and across parts of Texas. B21 watersheds were seen mostly in the central part of the eastern U.S. Watersheds belonging to class C1 occurred along the north-western coast of and Class C21 watersheds were found along the Appalachians and in the northeastern U.S. Streams in Class C22 did not have any specific regional structure and were distributed in different regions within the northern U.S. [40].

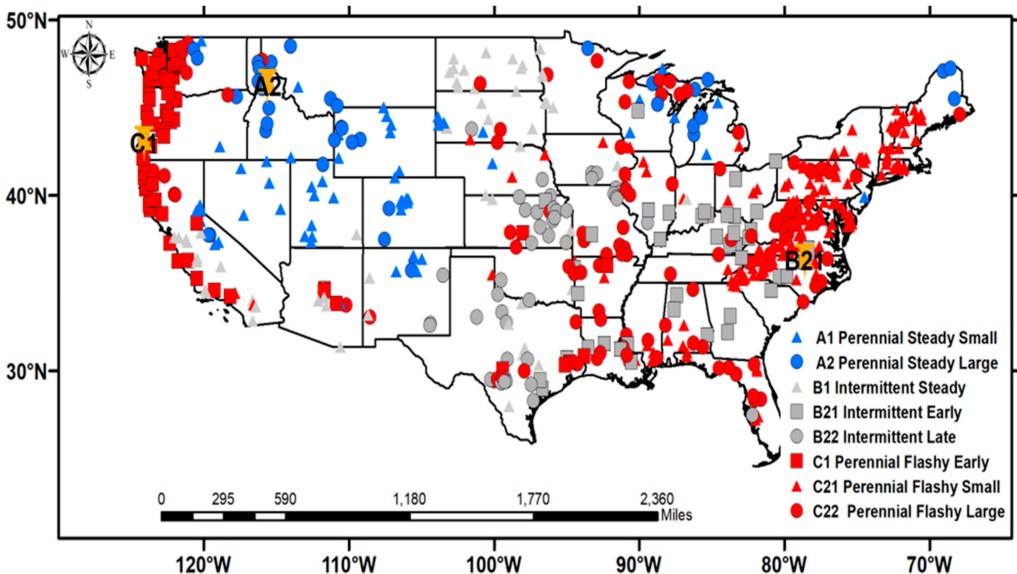

**Figure 2.** Spatial distribution of eight watershed classes across the US. The pushpin symbol indicates the study sites selected for observed and simulated streamflow comparison.

## 3. Results

### 3.1. Relationship between Soil Moisture, Precipitation and Streamflow

Lag correlation between soil moisture, precipitation, and streamflow was estimated to examine the spatial variability of the relationship between those variables as well as to investigate the potential of soil moisture for streamflow forecasting. Prior to the correlation analysis, the normality of the streamflow data was checked by using the Shapiro-Wilk test, and the $p$-values for most of the watersheds streamflow are less than the 5% significance value indicating the non-normality of original streamflow data (Table 1). In addition, the modified Mann-Kendall $p$-value is higher than 0.05 in most of the watersheds, implying no significant trend in the streamflow data.

**Table 1.** Normality and trend analysis test results for the streamflow data for eight watershed classes.

| Watersheds | Shapiro-Wilk Test Results (Median) for Original Streamflow | Shapiro-Wilk Test Results (Median) for Transformed Streamflow | Modified Man-Kendall Test Results(Median) for Original Streamflow |
|:---:|:---:|:---:|:---:|
| A1 | $1.07 \times 10^{-14}$ | 0.03 | 0.40 |
| A2 | $6.96 \times 10^{-12}$ | 0.29 | 0.36 |
| B1 | $1.35 \times 10^{-16}$ | 0.06 | 0.47 |
| B21 | $7.53 \times 10^{-11}$ | 0.48 | 0.34 |
| B22 | $3.95 \times 10^{-15}$ | 0.27 | 0.34 |
| C1 | $1.07 \times 10^{-7}$ | 0.21 | 0.27 |
| C21 | $8.42 \times 10^{-9}$ | 0.38 | 0.29 |
| C22 | $7.52 \times 10^{-11}$ | 0.33 | 0.34 |

In general, soil moisture and precipitation have a strong association with streamflow in most of the watersheds (Figures 3–5), indicating that changes in precipitation and soil moisture lead to streamflow changing. Results suggest that watersheds located in the Pacific Northwest and eastern part of the U.S. exhibited a higher correlation than those found in the Midwest. Watersheds located in the Midwest are characterized by intermittent streamflow regimes where the variability of the streamflow is much more significant compared to watersheds located in the Pacific Northwest, thus has lower correlation values. Our correlation analysis reveals varying strength of the correlation with watershed class and lag times (Figures 3–5). In the perennial steady watershed class (A1 and A2), the correlation coefficient

between precipitation and streamflow are mostly positive and increases with lag time, indicating that same and previous season precipitation and soil moisture have higher influence on the streamflow (Figure 6). The previous month's precipitation and soil moisture have higher influence compared to the same month's precipitation. As these watersheds are located in high mountainous regions where snowfall accounts for a significant amount of precipitation that accumulated during winter and release during spring. The highest correlation for the A1 and A2 watersheds are found for the lag 2 and lag 3 months, indicating that A2 watersheds exhibit longer precipitation and soil moisture memory compared to the A1 watersheds. As the A1 watershed class are located near the lower latitude and little snow accumulation results in shorter memory.

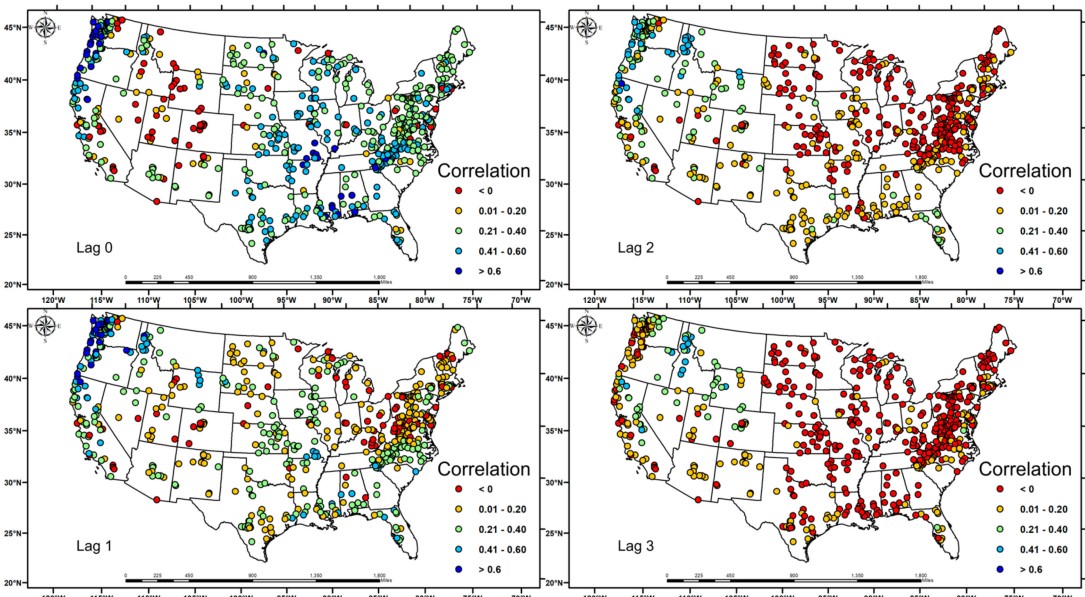

**Figure 3.** Spatial variation of correlation coefficients of precipitation-streamflow for different lag months.

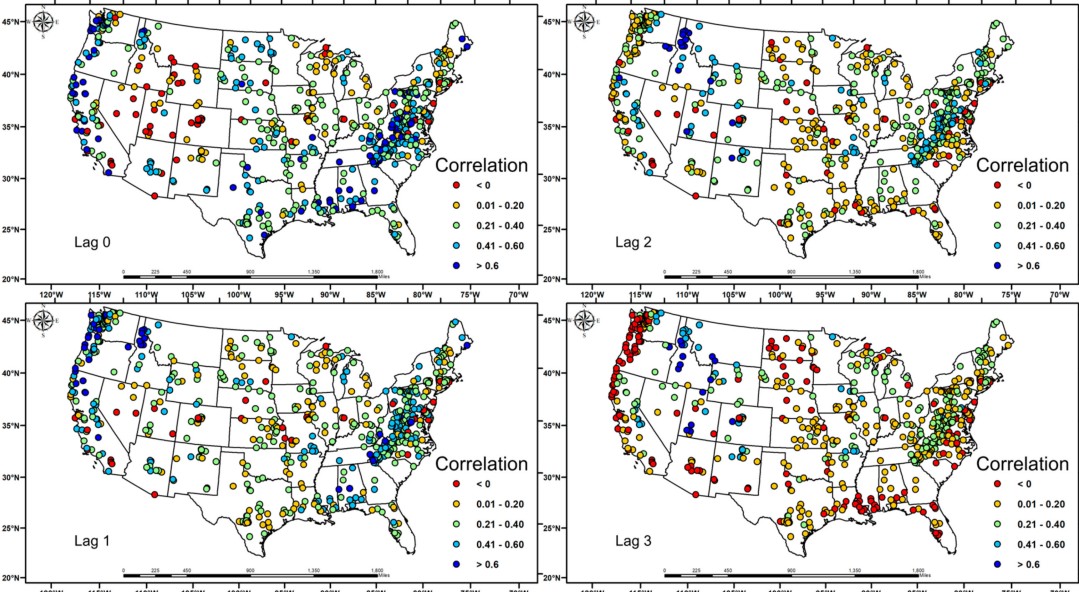

**Figure 4.** Spatial variation of correlation coefficients of surface soil moisture-streamflow for different lag months.

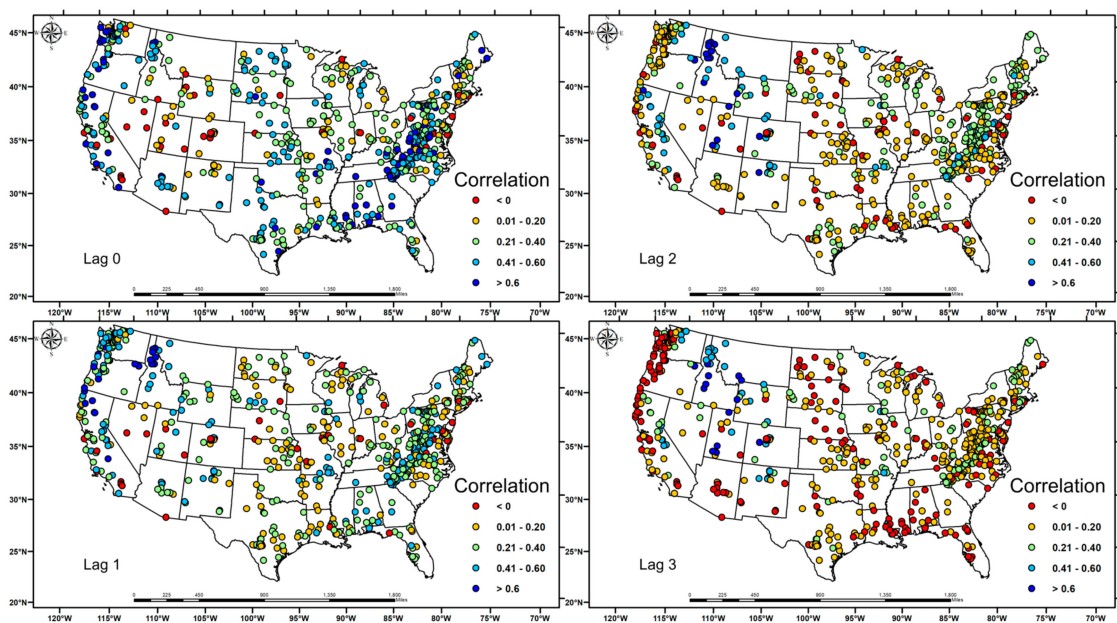

**Figure 5.** Spatial variation of correlation coefficients of root-zone soil moisture for different lag months.

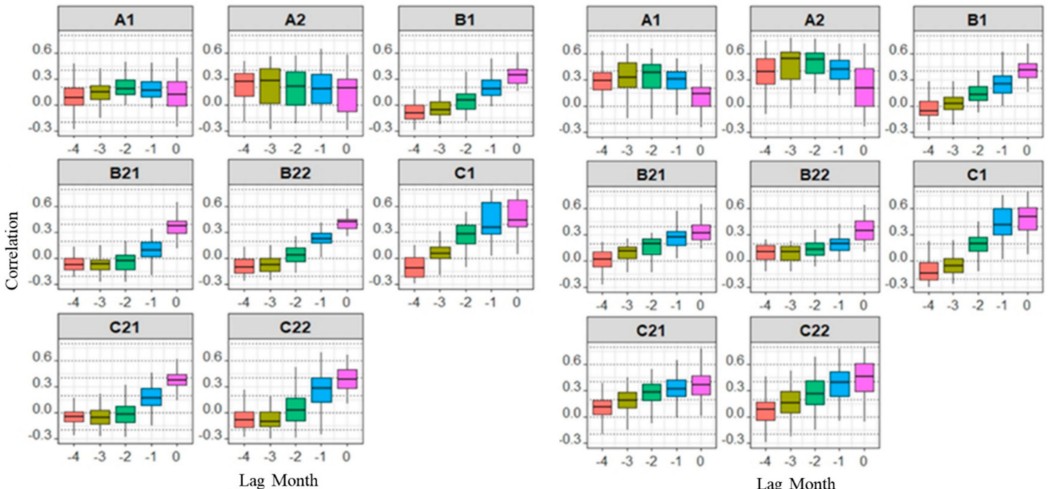

**Figure 6.** Box plots of lag correlations between precipitation and streamflow (**left**) and SMOS- based root zone soil moisture and streamflow (**right**) for eight watershed classes. The lower and upper ends of the box represent the first and third quartiles, respectively, and the whiskers extend to the extreme value within 1.5 IQR (interquartile range) from the box ends.

The watershed class with a drier condition (watershed class B) experiences weak coupling between soil moisture and streamflow among all watershed classes (Figures 4 and 5). The highest correlating is found between streamflow and precipitation at lag 0, which implies that the majority of the precipitation that falls within the watershed transforms into runoff immediately. The correlation coefficient decreases with higher lag months, indicating that little influence of prior month precipitation and soil moisture for the streamflow forecasting. These watersheds are mostly located near the low elevation of the basin, implying that precipitation occurs mostly as rain or as snowfall that melts soon after it occurs. Among all watershed classes, the watershed class (C1) with the highest precipitation and soil moisture exhibits the highest coupling of soil moisture and streamflow with the mean correlation values of 0.65 at lag 0 month. The watershed in the C1 watershed class, although located in the western coastal of the region, and runoff in those watersheds is caused predominantly by rainfall, and the response of the streamflow to rainfall is short.

Our results indicate that the surface and sub-surface soil moisture has a higher association with streamflow compared to the precipitation. For example, in the C1 watersheds, the median correlation value between precipitation and streamflow is 0.45 and 0.6 for the surface soil moisture-streamflow. This agrees with a previous study showing relatively higher coherence of soil moisture with streamflow [62]. Root-zone soil moisture and streamflow also exhibit strong correlations in most of the watershed and similar spatial patterns as the surface soil moisture-streamflow. However, the average correlation between root-zone soil moisture and streamflow (0.38) is slightly higher than surface soil moisture-streamflow (0.36). Previous studies suggested that precipitation can be used as a predictor of streamflow, with a lag of one to several months [63,64]. Our study also demonstrates the utility of remotely sensed soil moisture data as a predictor for streamflow.

### 3.2. Streamflow Forecasting

First, we present the intermediate results for the ARIMA and SVR model for a sample watershed, and findings pertaining to 601 watersheds were summarized later. The ACF and PACF plot (Figure 7) show higher auto-correlation values at lag 1, 2, and multiples of 12, implying the non-stationary of the streamflow data. The modified Man-Kendall (0.85) and Mann-Whitney *p*-values (0.35) are also higher than the corresponding significance level, indicating little trend and jumping in the streamflow data. The time series for the ARIMA should have a constant mean, variance, and autocorrelation correlation with time. However, both the ACF and PACF plot show significant value at lags 12, which indicates seasonality influence in the streamflow data. Therefore, the seasonality of the streamflow was removed by applying 12-month differencing on the original streamflow data, and the ACF and PACF were estimated to get the value of AR and MA part, as shown in Figure 8.

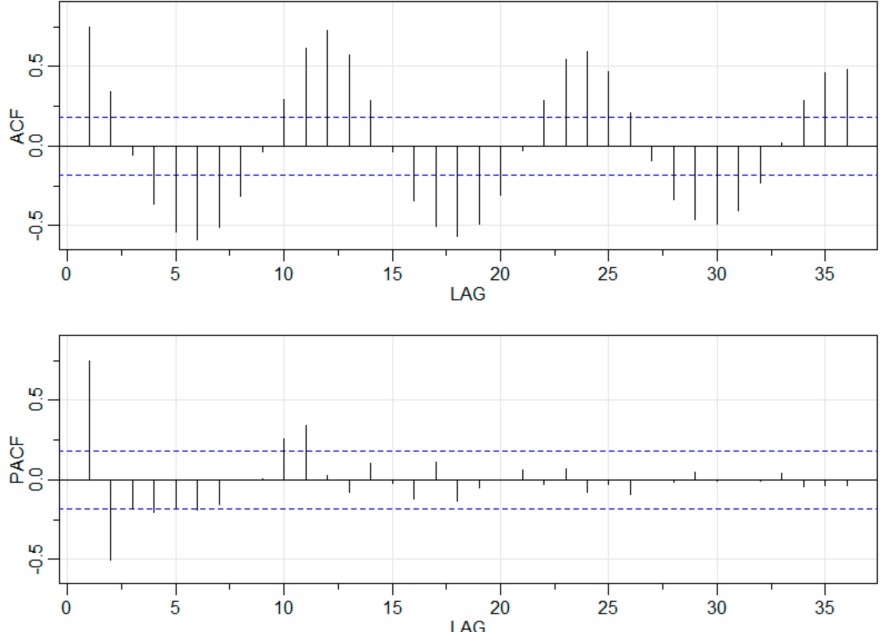

**Figure 7.** The ACF and PACF for monthly streamflow data.

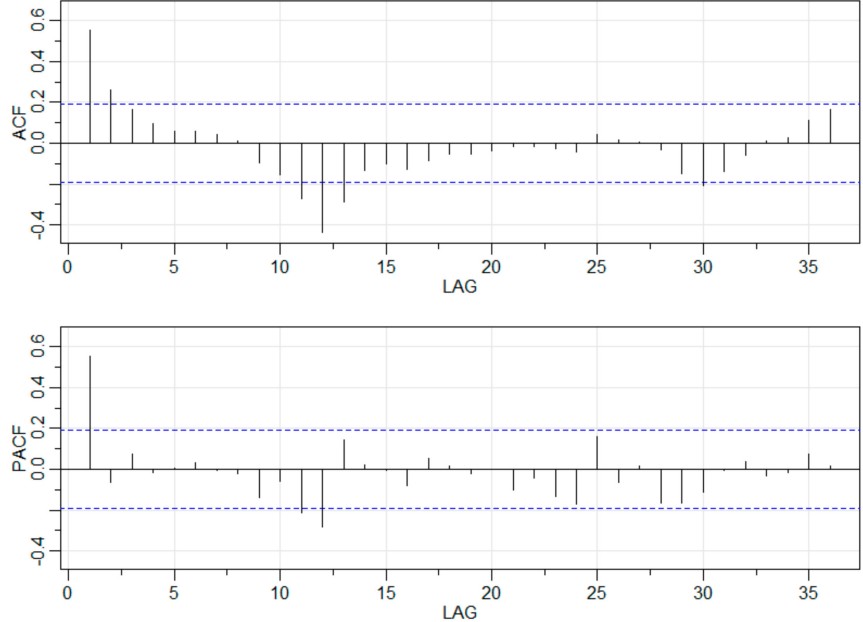

**Figure 8.** The ACF and PACF for transformed streamflow data showing auto and partial correlation for different lag months.

The PACF of transformed data shows the spikes at seasonal lags and only one significant peak at the non-seasonal lags. Based on the ACF and PACF plot, a different combination of the ARIMA models were developed, and the best ARIMA model was identified by using the minimum AIC values, as shown in Table 2.

**Table 2.** Values of the Akaike information criterion (AIC) for suggested ARIMA models.

| Model | AIC | Model | AIC |
|---|---|---|---|
| ARIMA(1,0,0)(1,1,0) [12] | 124.1372 | ARIMA(1,0,1)(1,1,0) [12] | 124.9614 |
| ARIMA(0,0,1)(0,1,1) [12] | 114.1933 | ARIMA(1,0,1)(1,1,2) [12] | 116.8087 |
| ARIMA(0,0,1)(0,1,0) [12] | 150.2422 | *ARIMA(1,0,0)(0,1,1)* [12] | *112.348* |
| ARIMA(0,0,1)(1,1,1) [12] | 116.1038 | ARIMA(1,0,0)(0,1,0) [12] | 146.962 |
| ARIMA(0,0,1)(0,1,2) [12] | 116.0554 | ARIMA(1,0,0)(1,1,1) [12] | 114.3932 |
| ARIMA(0,0,1)(1,1,0) [12] | 126.9248 | ARIMA(1,0,0)(0,1,2) [12] | 114.3695 |
| ARIMA(0,0,0)(0,1,1) [12] | 149.5471 | ARIMA(1,0,0)(1,1,2) [12] | 116.5605 |
| ARIMA(1,0,1)(0,1,1) [12] | 112.472 | ARIMA(2,0,0)(0,1,1) [12] | 112.8223 |
| ARIMA(1,0,1)(0,1,0) [12] | 147.5482 | ARIMA(2,0,1)(0,1,1) [12] | 113.9666 |
| ARIMA(1,0,1)(1,1,1) [12] | 114.5846 | ARIMA(2,0,0)(0,1,1) [12] | 112.8223 |
| ARIMA(1,0,1)(0,1,2) [12] | 114.569 | ARIMA(2,0,1)(0,1,1) [12] | 113.9666 |

The residuals for the fitted model (Figure 9) are normally distributed, and autocorrelation values are not significant, indicating that the residuals from the best model are white noise. The ACF and PACF plot for the sample watershed show that streamflow at lag one month has a significant correlation with monthly streamflow; therefore, previous month streamflow was considered as input for the $Q_{svr1}$ model. The cross-correlation between precipitation, root-zone soil moisture, and streamflow was found significant at lag one month; therefore, previous month precipitation and soil moisture were used as additional inputs in the $Q_{svr3}$ model. The grid search method evaluated the performance of the model with different combinations of the parameters, and the model with the lowest error was selected as the

best model. Similar to the sample watershed, the ARIMA and SVR models were developed for 601 watersheds, and the model evaluation results are summarized in Figure 10.

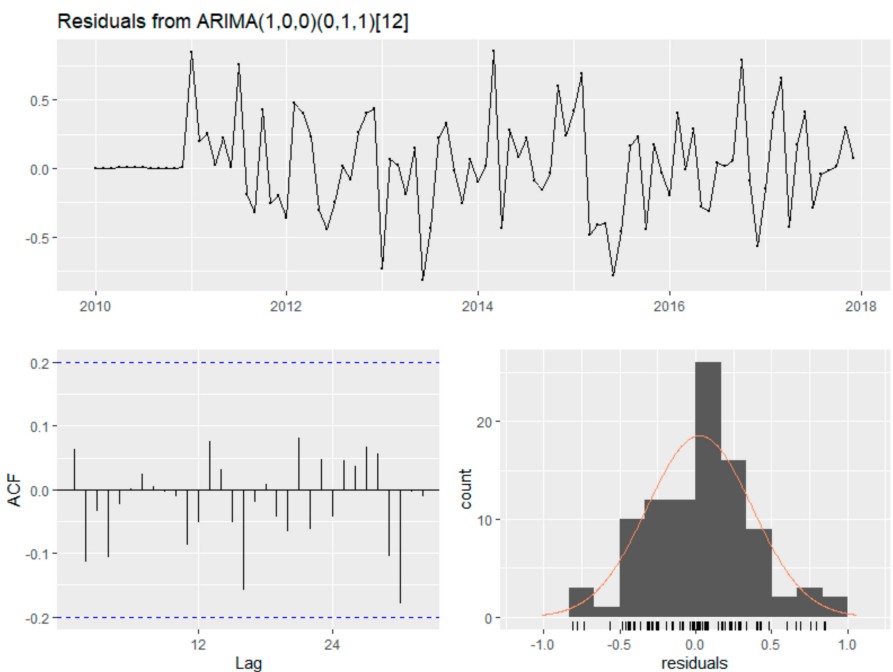

**Figure 9.** Residuals analysis for the ARIMA model.

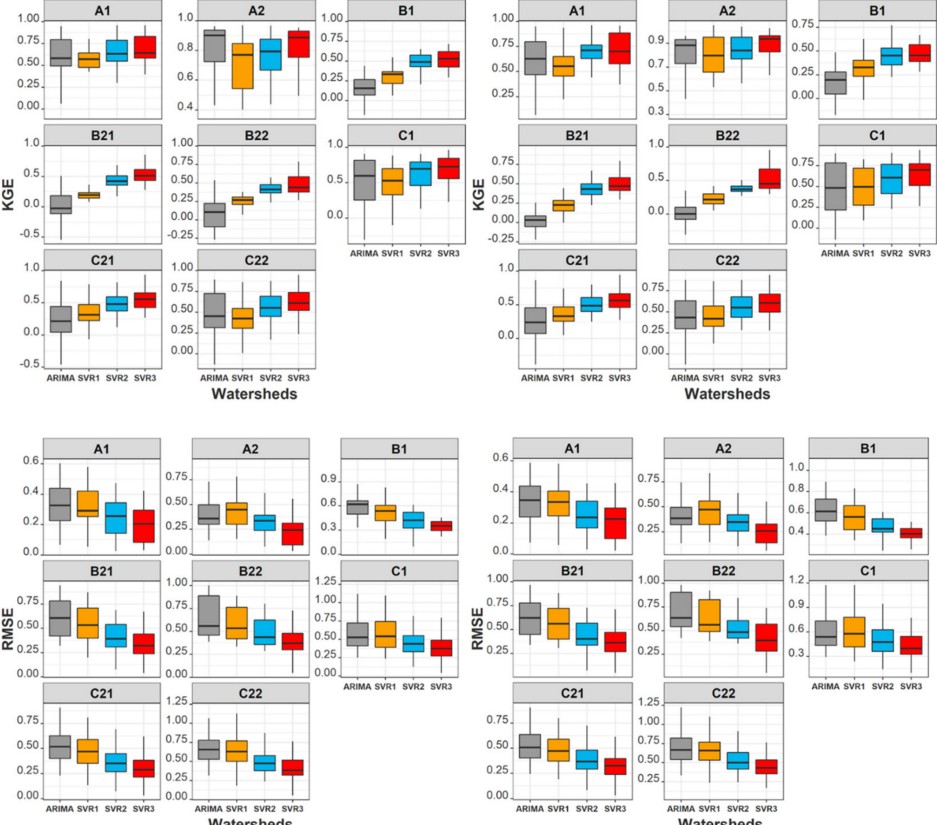

**Figure 10.** Kling-Gupta efficiency (KGE) (**top**) and root mean square error (RMSE) (**bottom**) boxplots of the $Q_{arima}$, $Q_{svr1}$, $Q_{svr2}$ and $Q_{svr3}$ models during calibration (**left**) and validation (**right**) period for the eight watershed classes.

The performance of the $Q_{arima}$ and $Q_{svr1}$ models indicate that watersheds located in the Pacific Northwest, such as C watershed class, exhibit better predictability than those found in the Midwest, such as watershed class B1 (Figure 10). Most of the watersheds class B are located in the dryer areas where occasional rainfalls create high flows that deviate significantly from the watershed's nominal flows, resulting in a more erratic flow regime and, thus, is less predictable. This pattern is generally in line with previous studies. Patil and Stieglitz et al. (2012) found that high predictability watersheds are bounded to the Cascade Mountains in the Pacific and Northwest Appalachian Mountains in the eastern US. In contrast, low predictability catchments are found mostly in the drier regions west of the Mississippi River [63,65]. A higher KGE and lower RMSE values are observed in the A watershed class compared to the C watershed class, which suggests that perennial steady watersheds tend to exhibit better predictably compared to perennial flashy watersheds due to lower variability in the streamflow. The $Q_{svr1}$ model outperforms the $Q_{arima}$ model in four watershed classes; the improved performance of the SVR model likely reflects its ability to capture the nonlinear and complex features of the streamflow process.

However, the ARIMA modeling approach reveals slightly better performance for the perennial steady watershed class. For example, the median KGE values for the perennial steady large watershed class during the training period is 0.9, which is around 25% higher compared to the $Q_{svr1}$ model. The $Q_{arima}$ model has the lowest KGE and highest RMSE values in the B watershed class. The streamflow in the B watershed class is intermittent, which has constant or zero flow during the dry period and only flows during the rainy season. Hence, streamflow in those watersheds are highly skewed, non-stationary, non-linear, and difficult to forecast using the ARIMA model. The model performance was also evaluated for the training and testing period to avoid the risk of overfitting. In general, the training and testing results were satisfactory and varied with the watershed class. For example, the performance of the $Q_{arima}$ model is slightly better in the A1 watershed, and it is slightly worse in the C1 watershed class during the training period as compared to the testing phase.

The $Q_{svr2}$ model that considered antecedent streamflow and precipitation as predictors shows an increase in model performance compared to the $Q_{svr1}$ model, and the magnitude of the improvement varies with different watershed classes. The median improvement in KGE values for including antecedent precipitation ranges from 0.06 to 0.24, and the greatest increase occurs in the B watershed class. The median improvements in RMSE value for including antecedent precipitation are about 0.1, 0.12, and 0.13 for the A, B, and C watershed classes, respectively, during the calibration period. The improvement in the model performance using antecedent precipitation and streamflow generally agrees with the results of other studies [66–68]. For example, Sivapragasam et al. (2007) applied genetic, and ANN to forecast streamflow using antecedent rainfall and streamflow and concluded that models with rainfall and streamflow made a more accurate forecast than those with only streamflow input [69]. However, the magnitude of improvement in the streamflow forecast models are not directly comparable, due to differences in lead times, the number of watersheds, and forecasting methods.

In addition to antecedent streamflow and precipitation observations, we investigated to what degree the inclusion of soil moisture data improves the streamflow forecasting capability. Inclusion of antecedent root-zone soil moisture in the model $Q_{svr3}$ increases KGE and decreases RMSE in most of the watershed class, as shown in Figure 10. For example, streamflow forecast using soil moisture data increases median KGE value by 22% and 14% compared to the $Q_{svr2}$ models, in the B21 and C22 watershed class, respectively, during the calibration period. The $Q_{svr3}$ model has the highest KGE (0.83 and 0.89), as compared to other SVR models, in calibration and validation period, respectively in the A2 watershed. The improvement of the KGE values was found to be slightly higher for the drier watershed compared to the wetter watershed class and consistent with the findings of Berg and Mulroy et al. (2006) who suggested that initial soil moisture conditions were less important in wetter, more snow-dominated watersheds [70]. The improvements in forecasted accuracy using soil moisture observations is consistent with other studies. For example, Harpold et al. (2017) showed that including soil moisture observations improved statistical streamflow forecast accuracy at 12 watersheds over in

Utah and California [71]. Maurer and Lettenmier et al. (2003) develop a multiple regression model to represent the joint contributions from soil moisture initialization and seasonal climate forecasts in the Mississippi River basin and found that soil moisture controls streamflow predictability for lead times of 1–2 months [7]. Similarly, Abdullah et al. (2019) incorporated antecedent soil moisture into forecasting streamflow volumes within the North Platte River Basin, Colorado/Wyoming (USA), and there result indicated better streamflow prediction when antecedent soil moisture used as an additional predictor in the forecasting model [39].

Figure 11 shows the predicted and observed streamflow for three selected watersheds from each watershed class. The A2 watershed has a well-defined streamflow regime, simulated streamflow shows better agreement with observed streamflow in $Q_{svr3}$, capturing the potential of soil moisture for improved streamflow prediction. On the contrary, inconsistencies between modeled and observed streamflow are observed for the B21 watershed. The observed streamflow is consistently higher than the simulated streamflow, indicating little influence of antecedent streamflow, precipitation and soil moisture in future streamflow conditions. The $Q_{svr3}$ simulated streamflow of the C1 watershed also shows better agreement with the observed streamflow.

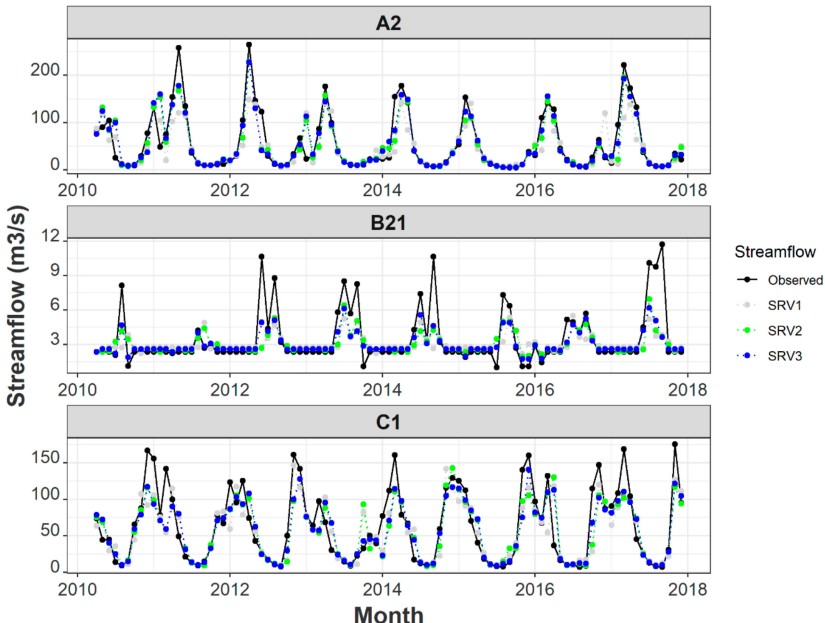

**Figure 11.** Comparison of observed and modeled streamflow for the large steady perennial (**A2**), early intermittent (**B21**), and early flashy perennial (**C1**) watersheds.

## 4. Discussion

Understanding the relationship between soil moisture, precipitation, and streamflow is important from both theoretical and practical perspectives. In this study, we evaluated streamflow response to changes in the precipitation and soil moisture among 601 different watersheds. Different watersheds have a wide range of climate and physiographic properties which influence the hydrologic regime; therefore we summarize the response of the 601 watersheds using eight classes, which enables us to assess and categorize the precipitation, soil moisture, and streamflow relationship [59]. The spatial pattern of the streamflow response appears to be consistent with the influence of topography and climate and not geographically distinct as some regions contain multiple numbers of watershed classes. We found some classes (e.g., class C21, B22) have more variability, possibly due to the larger number of watersheds within the class but can also be due to the broader diversity of physiographic, anthropogenic, or climatic factors in those watershed classes.

Our study suggests a variability of correlation among watershed classes determined by watershed heterogeneity along with the hydroclimatic process. For example, the watershed class with higher

precipitation (e.g., C1) showed a higher coherence between soil moisture and streamflow. Similarly, the watershed class with higher snow amounts showed an increase of correlation between soil moisture and streamflow with higher lag among all of the watershed classes. A comparison of streamflow and soil moisture coupling among different watershed classes provides a general indicator of the sensitivity of the streamflow of the watershed to precipitation and soil moisture changes. This approach emphasizes the role of antecedent soil moisture in the hydrologic system. To this end, understanding of the streamflow-climate interaction and isolating the soil moisture influence on streamflow could be used for improving the estimation of streamflow when modeling options are limited.

In general, in situ soil moisture provided a relatively accurate representation of soil moisture conditions. However, such observations are sparse, and providing limited information about the spatial coverage and variability [72,73]. In this study, we demonstrated the utility of remotely sensed soil moisture as an alternative to in situ measurements for streamflow forecasting. Our results indicate that remotely-sensed soil moisture leads to better streamflow forecasting in most of the studies watersheds and contributed to better decision-making in various areas of water policy and management.

Using the Google Earth engine as data processing platform reduces the technical challenges to process the data. The GEE platform provides high-performance computing infrastructure, thus speeding up the process of large volume geo-spatial and time series data considerably as compared to desktop computing. This has the potential to save time and improve scientists' ability to perform reproducible analysis as all input data and codes are available through GEE. In addition, the user does not need to install any additional software, which reduces the compatibility limitations, thus increases the data and tool accessibility and usability. It is relevant to note that there are limitations associated with assessing the streamflow responses to precipitation and soil moisture changes. This study used SMOS based soil moisture and PRISM precipitation data to quantify the relationship with streamflow. However, multiple sources of soil moisture and precipitation data are available, and the choice of data source can have an impact on the streamflow-climate relationship. Monthly precipitation data were used for evaluating the changes in streamflow; however, it is important to consider that streamflow can also be influenced by the intensity and concentration of the precipitation. In this study, antecedent streamflow, precipitation, and soil moisture were used as potential predictors for streamflow forecasting. However, other meteorological variables such as temperature, humidity, and wind speed also affect the streamflow. In future research, these factors will be taken into consideration as potential inputs for streamflow forecasting. This study used two data-driven models for streamflow forecasting; however, there are multiple statistical and physically models available, and the choice of the model might have an impact on the streamflow forecasting.

## 5. Conclusions

The overall goal of this manuscript was to explore the association among the precipitation, soil moisture, and streamflow and to evaluate the potential of satellite-based soil moisture products in streamflow forecasting models over a wide range of watersheds in the United States using GEE. A significant correlation between precipitation, soil moisture, and streamflow was found in most of the watersheds; however, the association varies with different watershed classes and lag times. Watersheds characterized as perennial steady showed higher streamflow forecasting capability compared to intermittent watersheds. The spatial pattern of the correlation values reflects that basins located in the Pacific Northwest and the eastern U.S. generally have a higher correlation between soil moisture and streamflow than those located in the Midwest. Strong positive soil moisture-streamflow relationship was found, in particular where the soil moisture precedes or is concurrent with the streamflow. The highest correlations were observed in the snow-dominated watersheds, where soil moisture precedes streamflow by 2–3 months. Our streamflow forecasting results indicated that SVR outperformed or performed as well as the ARIMA model in most of the watersheds. The SVR and ARMA model performance varied considerably among the different watershed classes, where perennial watersheds exhibited better predictability compare to the intermittent watersheds. The SVR models with antecedent

precipitation and streamflow as predictors performed better forecasting, regardless of the watershed class than that with only streamflow input. We also incorporated root-zone soil moisture in the streamflow forecasting model because of its higher correlation with streamflow. The forecasted model showed that the inclusion of satellite base root-zone soil moisture products improved the streamflow in most of the watersheds. These findings compliment previous studies mentioned in Section 1, and contribute to a growing understanding of the complicated relationship between soil moisture, precipitation, and streamflow. We quantify these relationships over several types of watersheds, isolating the impact of climate conditions, precipitation, and soil moisture on streamflow response. In addition, it is envisaged that this study will further the application of satellite-based soil moisture and precipitation for improving streamflow modeling and forecasting.

**Supplementary Materials:** The supplementary (SMOS soil moisture data sets) materials are available online at http://www.mdpi.com/2073-4441/12/5/1371/s1, and available at https://explorer.earthengine.google.com/#detail/NASA_USDA%2FHSL%2Fsoil_moisture. PRISM climate data sets link: https://developers.google.com/earth-engine/datasets/catalog/OREGONSTATE_PRISM_AN81m.

**Author Contributions:** N.S., I.M., and J.B. designed the work; N.S. undertook the data analysis. All authors contributed equally to the final version of the manuscript.

**Funding:** This work is supported by the NASA Applied Sciences Program.

**Conflicts of Interest:** The Authors declare no conflict of interest.

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
