# Peer review of "Exploring Spatiotemporal Relations between Soil Moisture, Precipitation, and Streamflow for a Large Set of Watersheds Using Google Earth Engine"

_water, doi:10.3390/w12051371_

Round 1

Reviewer 1 Report

Summary

The study investigates hydroclimatic variability and monthly streamflow forecasting over CONUS. Based on a large and diverse sample of catchments, the study shows the complex interplay between precipitation, soil moisture, and streamflow. Furthermore, it highlights the potential of the NASA USDA global soil moisture product for streamflow forecasting: Based on the findings from the explorative analysis, regression models are developed and evaluated with respect to the accuracy in forecasting monthly streamflows. In general, the model that has access to soil moisture information performs best.

General comments

The study uses data at the monthly time scale -- this should be more stressed already in the abstract, since the time scale is fundamental, in particular when it comes to the 'forecasting part'. The literature review in the introduction could benefit by following more closely the actual focus of the study, i.e. hydroclimatic variability and streamflow forecasting over CONUS at the monthly time scale. This would also provide references that could be taken up in the discussion of the results. Please revise the results-section with respect to redundant information and stylistic/grammatical issues. Even as a non-native English speaker, I can spot frequently mistakes. In addition, one could argue that comparing the results with other studies would be more appropriate in the discussion-section.

Specific comments

L88-L94: Either include this information in the paragraph above or skip it -- in my opinion this information is at the wrong place. In addition: Why do you note cite these "very few studies"? That could be interesting, in particular to compare with your results.

L98-105: These sentences could be skipped without any loss of information, as the methodology follows immediately below.

L109-127: Please note the time period of your data and the resulting sample size. Are data gaps present in the streamflow time series?

L148: What do you mean by variable independence? ARIMA models handle the correlation of the variable at different time lags in a smart way (essentially, this is what they are supposed to do). Moreover, I have the impression that you are using an AR-like model (i.e. OLS linear regression with lagged predictors); if so, it would be more appropriate to name it accordingly. Otherwise, briefly mention how do you approach the model identification (i.e. the selection of the AR, I and MA part and the parameter estimation procedure).

L149: The SVM is a linear model at its heart, but can incorporate non-linear mappings via basis expansions of the predictors (usually done via the kernel trick). Please specify how do you technically account for potential non-linearities in the data set.

L152: Please note your model validation strategy -- e.g. does it involve a cross-validation? If the validation is based on in-sample predictions, this must be emphasized.

L157: Why do you use bias to evaluate the forecast performance? Statistically based approaches should fairly reproduce the mean value of the predictand, thus are approximately unbiased in this respect. Bias is much more important when it comes to simulation models. I suggest to skip the bias quantity or replace it by a more advanced NSE, where you plug in for example the monthly streamflow climatology or even the ARIMA model in the denominator of the NSE definition (which is then usually referred to as the mean squared error skill score/MSESS).

L193: I suggest to remove "streamflow" as it is discussed in the paragraph below; in addition, the statement is only partly correct when including streamflow due to snow and ice (also reference [28] deals not deal with streamflow).

L224: Why do we have eight pushpin symbols here? In Fig. 7 only three are actually used.

L253: "...implies that majority of the precipitation that falls within the watershed transform into runoff immediately" -- if so, you would observe the highest correlation between streamflow and lag 0-precipitation (and not soil moisture).

L335: What causes the streamflow in catchment B21 to be almost constant during several consecutive time steps? Is this a strongly regulated catchment?

L367-369: This information would be more appropriate in the introduction.

L380: "It is relevant to note" -- line break here? Has no link with GEE.

Typos and grammar

L37: efficiency?

L59: is this sentence correct? "... soil moisture was identified as a ..."?

L80: heterogeneity

L136: deseasonalised?

L206: "compared" instead of "compare" -- this mistake follows frequently below

L208: "and travel times of water to from runoff sources" -- why not simply "and water travel times"?

L256: increasing lag time?

L305: predictability

L310: to the using ...??

L398: reflect

L399: have

L401: precedes

L406: its

Reviewer 2 Report

All the comments are in the joint file

Reviewer 3 Report

Overall the paper is very well designed, thought out, and presented.  There are only two areas where I think that there can be improvement.

Lines 109 – 110 “This study was conducted over total of 601 watersheds across U.S. which are least effected by human influence on streamflow” How were these 601 watersheds determined to be “least effected”? What criteria were used to determine which watersheds were effected and which not? Is that a binary determination or is there a continuum of values that are used to assess degrees of “effected-ness”?  If the latter, what is the threshold on that continuum What is meant by “over total”? Is the number 601 or something else? How many other, more effected watersheds are there? This should stated.  Is 601 are large portion of all watersheds or a small portion?

Line 133 - 135 “Next, the association between soil moisture, precipitation, and streamflow were estimated using Spearman rank correlation for different lag times.” The Spearman rank correlation is used for non-normally distributed data sets. Is this data non-normally distributed? If this data is indeed non-normally distributed, this should be stated as such. More importantly there should be some sort metric to assess normality and how far this data is from this.  Something like the Kolmogorov-Smirnov test or Shapiro-Wilk test would be useful to assess the normality of the data.

Reviewer 4 Report

The authors studied the relationship between precipitation, soil moisture, and streamflow of a large watershed using Google Earth Engine (GEE). This work is important for hydroclimatic variable as it provides the foundation for the further researches on mitigation of the impacts of floods on agriculture and other human activities. It is clear that the nature of the hydrological variables in time series modeling has great impact on the predicted output and should thus be considered in the modeling procedure. My main concern is that how you selected model input parameters? The authors considered t-1 and t-2 for building three tested input combinations. A key question regarding nonlinear modeling that remains unanswered in the manuscript is whether the best input combination is selected. How the best input combination was selected? More details about it should be added in the manuscript. I recommend publication of this study with some major revisions .

  • Why using a nonlinear model (SVM)? Not other algorithms: ELM, ANFIS? Additional information should be added in manuscript.
  • The seasonal effect should be verified by ACF and PACF graphs.
  • The authors need to include a good literature survey on time series modeling; they just focus on few studies (P.2), there is no detail about application of these methods in some practical cases. I would like a clear discussion on the literature versus the unique contribution of the paper. The authors may want to consider papers describing time series modeling to real-world problems. Among other contributions the authors should refer to the following:

Implementation of Univariate Paradigm for Streamflow Simulation Using Hybrid Data-Driven Model: Case Study in Tropical Region, IEEE Access, Vol. 7, pp. 74471-74481., doi: 10.1109/ACCESS.2019.2920916.

Novel approach for streamflow forecasting using a hybrid ANFIS-FFa model, Journal of Hydrology, Vol. 554, pp. 263-276. doi: 10.1016/j.jhydrol.2017.09.007.

Evaluation of preprocessing techniques for improving the accuracy of stochastic rainfall forecast models, International Journal of Environmental Science and Technology, doi: 10.1007/s13762-019-02361.

New insights into Soil Temperature Time Series Modeling: Linear or Nonlinear? Theoretical and Applied Climatology, Vol. 135(3-4), pp. 1157-1177. doi: 10.1007/s00704-018-2436-2

A reliable linear stochastic daily soil temperature forecast model, Soil & Tillage Research, Vol. 189, pp. 73–87. doi: 10.1016/j.still.2018.12.023.

A reliable linear method for modeling lake level fluctuations, Journal of Hydrology, Vol. 570, pp. 236-250. doi: 10.1016/j.jhydrol.2019.01.010.

Stochastic model stationarization by eliminating the periodic term and its effect on time series prediction, Journal of Hydrology, Vol. 547, pp. 348-364. DOI: 10.1016/j.jhydrol.2017.02.012.

Impact of Normalization and Input on ARMAX-ANN Model Performance in Suspended Sediment Load Prediction, Water Resources Management, Vol. 32, No. 3, pp. 845-863. doi: 10.1007/s11269-017-1842-z.

Forecasting Monthly Inflow with Extreme Seasonal Variation Using the Hybrid SARIMA-ANN Model, Stochastic Environmental Research and Risk Assessment, Vol 31, No. 8, pp. 1997-2010. DOI: 10.1007/s00477-016-1273-z.

  • SVM model parameters should be presented in details. The rationale on the choice of the particular set of parameters should be explained for all methods. What are the sensitivities of these parameters on the results? More details should be provided.
  • A sensitivity analysis should be carried out to define the most important input parameters. To get more information on the way the authors apply sensitivity analysis model
  • How the authors train and test AI approaches? It should be discussed in details.
  • In addition to the applied statistical criterions, the authors should use Bayesian Information Criterion (BIC) and Schwarz Bayesian Criterion (SBC) to define the best model.
  • Was any transformation applied to the data? Need to be detailed.
  • The authors must employ n-folds cross-validation or leave one out cross-validation to increase the cogency of the results.
  • I also wonder why the authors did not employ any feature selection algorithm to select most relevant features instead of performing these amount of model trials
  • The authors should explain why did not you compare the performance of proposed methods against stochastic methods, I strongly recommend that the method should be compared against more methods to prove it is superiority.
  • The authors have to investigate the linear models such as ARIMA in detail. The simple comparisons without physical restrictions between linear and nonlinear models cannot suggest the scientific background approaches.
  • Please add the detailed information for the research processes between low-order and high-order ARIMA models.
  • The normality results of various transformations using the Jarque-Bera, Doornick chi-squared and Anderson-Darling tests should be included in the manuscript. In the paper, the authors used these test to normalize data but they did not provide any its results that indicate the proof of normality.
  • The author should present details about the time series terms such as periodic term, jump etc.
  • The author should add more details about stationarization and normalization of time series.
  • The authors should mention how to eliminate the influence of serial correlation on the Mann-Kendall test.

Please, add all dataset values in the supplementary materials (all input paramters+results of these two methods), it is important for other studies which want to use your studies within an open access journal like Water.

Reviewer 5 Report

I think the authors carried out a large amount of work to explore relationship between moisture, precipitation, and streamflow for a large set of watersheds using Google Earth Engine. The paper is generally well-written in an understandable way, and the use of English is good. I recommend the manuscript for publication after the following  few  major changes:

  • What is the novelty of the proposed algorithms and its potential impacts, over other established Machine and deep algorithms e.g. machine learning techniques (QRF, RF, BART etc. Authors should explain on this aspect in the introduction section.
  • The accuracy of model is evaluated using BIAS metric which is indication only systematic error. Also could you please provide any random error statistics such as Root mean square error (RMSE). They you can properly justify the robustness of your model with systematic and random error.
  • Can you please provide modified Kling–Gupta efficiency (KGE) results instead of NSE .  KGE provides three distinct components representing the correlation, the bias, and a measure of relative variability in the simulated and observed values which will significantly verify your model performances.
  • Verification Methodology: The methodology section sections is very long. If this is standard method, can it be condensed?  You can provide a schematic flowchart which will present the big picture of this research work These could  be  necessary component of a manuscript. 

  • No detail information is provided on how the training and validation of the method is performed. Do you think these amount datasets are enough to construct machine learning model? How much dataset used for validation and testing? Please justify ?

  • There is also no information on avoiding overfitting. One of the challenges in data driven methods is overfitting (i.e. the method is so fine tuned to the training data, and has larger errors when applied to new datasets). I don’t see any discussion of this in the paper. Are there noticeable differences between the performance of the method during training and validation? I'd like to see whether the your could provide a reasonable model without much overfitting. This can be checked by dividing the data into training and testing subsets. The training and testing fitting errors should be comparably low.

Minor :

Could you please cite recent publications in introduction section which are completely missing in your paper. They found that the soil moisture is an important predictor to obtain a reliable reference rainfall product as well as stream flow estimates based on different machine learning or other non-parametric  algorithms

Zorzetto, E., & Marani, M. (2020). Extreme value metastatistical analysis of remotely sensed rainfall in ungauged areas: Spatial downscaling and error modelling. Advances in Water Resources135, 103483.

Kumar, A., Ramsankaran, R.A.A.J., Brocca, L. and Munoz-Arriola, F., 2019. A Machine Learning Approach for Improving Near-Real-Time Satellite-Based Rainfall Estimates by Integrating Soil Moisture. Remote Sensing, 11(19), p.2221

Camici, S., Crow, W.T. and Brocca, L., 2019. Recent advances in remote sensing of precipitation and soil moisture products for riverine flood prediction. Extreme Hydroclimatic Events and Multivariate Hazards in a Changing Environment: A Remote Sensing Approach, p.247.

Bhuiyan, M. A. E., Nikolopoulos, E. I., Anagnostou, E. N., Quintana-Seguí, P., and Barella-Ortiz, A.: A Nonparametric Statistical Technique for Combining Global Precipitation Datasets: Development and Hydrological Evaluation over the Iberian Peninsula, Hydrol. Earth Syst. Sci., https://doi.org/10.5194/hess2017-268 , 2018. 

Dorigo, Wouter, et al. "ESA CCI Soil Moisture for improved Earth system understanding: State-of-the art and future directions." Remote Sensing of Environment 203 (2017): 185-215.

Bhuiyan, M. A. E., E. I., Anagnostou, P.E. Kirstetter: A non-parametric statistical technique for modeling overland TMI (2A12) rainfall retrieval error. IEEE Geosci. Remote Sensing Letters, 14, 1898–1902, 2017.

 Hazra, A., Maggioni, V., Houser, P., Antil, H. and Noonan, M., 2019. A Monte Carlo-based multi-objective optimization approach to merge different precipitation estimates for land surface modeling. Journal of hydrology, 570, pp.454-462.

Round 2

Reviewer 2 Report

see joined file

Author Response

First of all, thank you for your comments and suggestions that allowed us to greatly improve the quality of the manuscript. We agree with all your comments, and we corrected point by point the manuscript accordingly. Please see attached responses. 

Reviewer 4 Report

Please increase font in Figure 5

Author Response

First of all, thank you for your comments and suggestions that allowed us to greatly improve the quality of the manuscript. We agree with your comment and have split the figure 5 into three figures.

Reviewer 5 Report

Overall, the paper is significantly improved and it will contribute to global water resources research.

Author Response

Thank you for your comments and suggestions that allowed us to greatly improve the quality of the manuscript. 

Round 3

Reviewer 2 Report

see attached file
